# Adolescents’ Sexual and Reproductive Healthcare-Seeking Behaviour and Service Utilisation in Plateau State, Nigeria

**DOI:** 10.3390/healthcare10020301

**Published:** 2022-02-04

**Authors:** Esther Awazzi Envuladu, Karlijn Massar, John de Wit

**Affiliations:** 1Department of Community Medicine, College of Health Sciences, University of Jos, Jos P.M.B 2084, Nigeria; 2Department of Interdisciplinary Social Science, Utrecht University, P.O. Box 80140, 3508 Utrecht, The Netherlands; j.dewit@uu.nl; 3Department of Work & Social Psychology, Maastricht University, P.O. Box 616, 6200 Maastricht, The Netherlands; karlijn.massar@maastrichtuniversity.nl

**Keywords:** adolescents, health-seeking behaviour, utilisation, sexual and reproductive health, Nigeria

## Abstract

The high rate of sexual and reproductive health (SRH) challenges among adolescents in Nigeria has been linked with the poor access to and utilisation of health facilities. This study explores the factors that influence the actual use and willingness to use SRH services among adolescents. Survey questionnaires were administered to 428 adolescents aged 18 to 19 years in six local government areas (LGAs) in Plateau State. The results showed that more than one-third of the participating adolescents were currently sexually active, slightly more than three-quarters (76.6%) had never visited health facilities for SRH issues, and more than half (56.0%) were not willing to visit a health facility if they ever had any SRH issues. The most frequent reason for the non-use of health care facilities for SRH issues by adolescents was a perceived lack of privacy and confidentiality (66.1%), followed by the perceived negative attitude of health care providers (68.2%). However, being sexually active was the only independent covariate of seeking SRH care from health facility (AOR = 005; CI = 0.01–0.49; *p* = 0.011), while awareness of HIV was a significant covariate of willingness to seek SRH care in a health facility in the future (AOR = 3.17, 95% CI = 1.50–6.70; *p* = 0.002). We concluded that the utilisation of SRH services and willingness to do so in the future was fairly limited among adolescents in this study. Therefore, there is a need to address the challenges of privacy and confidentiality and commencement of the health promotion of SRH for adolescents ahead of sexual initiation to alleviate the SRH challenges adolescents encounter when sexually active.

## 1. Introduction 

The high prevalence of sexually transmitted infections (STIs), HIV and unintended pregnancies among adolescents, referred to as teenage pregnancy, in sub-Saharan Africa, Nigeria inclusive, is associated with unprotected sexual activities and continues to be a great public health concern [1,2]. Condomless sex and a lack of contraceptive use among sexually active adolescents have been widely reported in many studies in Nigeria [3,4,5]. Among the adolescents who do report condom use during sex, many report incorrect and inconsistent use, which equally places them at risk of sexual and reproductive health (SRH) problems [6,7]. 

The high rate of unprotected sex among adolescents, especially those involved in risky sexual behaviour such as sex with multiple partners, has been associated with poor knowledge of the risk of exposure to SRH problems, such as STIs, HIV, and unintended pregnancy [8,9,10] and the lack of access to contraceptives among adolescents, as reported in many studies [11,12,13]. Importantly, however, some of the major reasons for low condom and contraceptive use are the lack of adolescent-centred sexual and reproductive healthcare services, and the unwillingness of adolescents to seek care in health facilities [10,14].

Health care facilities provide an important setting for adolescents to access accurate SRH preventive information, testing and treatment services. Unfortunately, adolescents’ utilisation of these facilities has remained low due to fear of being stigmatised, negative attitudes of health care providers and a lack of age-appropriate and adolescent-centred services [15,16]. For example, Odo et al. [17] report that, for adolescents in Enugu State, SRH services were perceived as financially inaccessible and not adolescent-friendly. In many African countries, religion and social structures create certain norms around ASRH, which in turn create barriers for adolescents using these services [15]. Rather than seek health care in health facilities, some adolescents in Nigeria have reported seeking SRH care from unprofessional providers such as medicine vendors, traditional healers, and others self-medicate by purchasing drugs without prescription against the backdrop of antibiotic abuse and resistance [18,19,20]. 

Promoting the appropriate use of SRH care is viewed as one of the key strategies in addressing the SRH challenges that adolescents face. However, there are still gaps in understanding the factors influencing the SRH-seeking behaviour of adolescents and why adolescents do not seek care in health facilities. This study explores the SRH-seeking behaviour of adolescents, the factors influencing health-seeking behaviour, and the willingness of adolescents to seek SRH care in health facilities in the future. The findings from this study can inform interventions that will provide responsive adolescent health services in Nigeria.

## 2. Methods

### 2.1. Study Design and Setting 

A cross-sectional survey was conducted among adolescents in six local government areas (LGAs) in Plateau State, located in the north–central region of Nigeria. These LGAs have both public and private health facilities that provide SRH services, which adolescents can access. All participants self-completed a questionnaire adapted from the WHO [21,22] and other studies, with slight modifications to contextualise the content to the Nigerian setting after pre-testing of the questionnaire. Information on socio-demographic characteristics, sexual activity, awareness of SRH issues, past health seeking for SRH issues, willingness to seek SRH care at health facilities in the future, and reasons not to seek SRH from health facilities were collected. 

### 2.2. Study Participants and Recruitment

Sample size was calculated using the Cochran formula (n_0_ = Z^2^pq/e^2^), where n is the minimum sample size, Z is the standard normal deviate which corresponds to a 95% confidence interval (1.96) and p is the proportion of adolescents seeking health care for SRH in health facilities. Here, we assumed it to be 50%, where q is the complementary probability (1 − *p*), and e is the 5% margin of error. The minimum sample size required was N = 384, but 428 adolescents participated in the study.

The inclusion of participants occurred in three stages. First, six LGAs were selected from seventeen LGAs by balloting. Next, eighteen wards (three from each of the six LGAs) were randomly selected. The research team then liaised with representatives of organised youth groups in each ward to identify households with adolescents aged 18 or 19 years old, mainly because they could give consent to participate in the study. Trained research assistants went from house to house to distribute the questionnaire and waited to retrieve the anonymous self-completed questionnaires from the adolescents, ensuring that the questionnaires were properly filled out for analysis.

### 2.3. Measures 

Socio-demographic characteristics assessed included gender, age, marital status, schooling status (in or out of school), and highest level of educational qualification. Whether participants were sexually active or not was assessed by asking if they were currently having sex. Awareness of SRH issues, STIs and HIV were assessed by directly asking if they had heard of sexual and reproductive health, sexually transmitted infections and HIV, respectively (yes/no; don’t know responses were recorded as no).

Health seeking for SRH was assessed by asking participants if they ever visited a health facility for SRH issues in the past (yes/no). Willingness to seek SRH care at a health facility was assessed by asking if they would be willing to visit a health facility in the future if they ever experienced any SRH issues (yes/no).

Reasons for non-utilisation of health facilities for SRH issues were assessed with an open-ended question. Responses were grouped into three themes: perceived lack of privacy and confidentiality, perceived cost of services and commodities, and negative attitude of health care providers. 

### 2.4. Data Analysis

Data were analysed using the IBM Statistical Package for Social Sciences (SPSS), version 23 (IBM Corp, Armonk, NY, USA). The socio-demographic characteristics, sexual activities, awareness of STIs and HIV were analysed as frequencies, and the level of utilisation of health facilities and willingness to use health facilities, including the reasons for non-utilisation of health facilities, were also analysed as frequencies.

The covariates of use of health facilities and willingness to use the health facilities for SRH problems were analysed using univariate and multivariate logistic regression. Here, the predisposing variables were the socio-demographic characteristics (age, gender, schooling status (either currently in school or out of school), educational status, etc.), sexual activity and awareness of SRH, STIs and HIV.

### 2.5. Ethical Approval

The Jos University Teaching Hospital (JUTH) Research Ethics Committee granted approval for this study. Permission was also obtained from the LGA authorities and leaders at the ward levels. All participants provided written informed consent before the commencement of the study.

## 3. Results

The study participants included both males and females (51.9% and 48.1%, respectively). Slightly more than half (52.3%) were 18 years old (47.7% were 19 years old). Few of the adolescents (5.8%) were married; 41.1% were out of school and the majority (74.1%) had a secondary education (See Table 1).

More than one-third (38.0%) of the adolescents were sexually active. About half (53.0%) were aware of SRH issues, nearly three-quarters (73.0%) were aware of STIs and the majority (85.0%) were aware of HIV. Slightly more than three-quarters (76.6%) had never visited a health facility for SRH issues, and more than half (56.0%) were not willing to visit a health facility if they ever had any SRH issues (See Table 2).

The most frequent reason for adolescents’ non-use of health care facilities for SRH issues was a perceived lack of privacy and confidentiality (66.1%), followed by a perceived negative attitude of health care providers (68.2%), and the cost of services and commodities (39.0%) (See Table 3).

Bivariate logistic regression analysis showed that age, marital status, schooling status, sexual activity, and awareness of SRH, STIs and HIV were significantly associated with the odds of seeking care for SRH in a health facility (*p* ˂ 0.05). Specifically, younger adolescents had reduced odds of seeking care in health facilities; the odds of health-seeking for SRH were about 2.3 times higher in the married compared to the unmarried; out of school adolescents had reduced odds of health seeking for SRH; those not sexually active had less odds of seeking health care in health facilities; the odds of health seeking for SRH in health facilities among those aware of SRH, STIs and HIV were 1.6 times, 3.1 times and 4.3 times, respectively, compared to those who were not aware. However, after controlling for the effects of all independent variables using a multivariate logistic regression, only being sexually active was independently associated with having sought SRH care from a health facility (AOR = 005; CI = 0.01–0.49; *p* = 0.011) (See Table 4).

Bivariate logistic regression analysis revealed that only awareness of HIV was significantly associated with willingness to seek healthcare in a health facility in the future. Adolescents who were aware of HIV had about three times more odds of future SRH health seeking in health facilities compared to those who were not aware of HIV (AOR = 2.76, 95% CI = 1.51–5.03). After adjusting for the effect of other independent variables, those with a higher educational status were significantly associated with a willingness to seek care in a health facility in the future (AOR = 1.34, *p*-value = 0.049). Additionally, adolescents who were aware of HIV had three times greater odds of willingness to seek SRH care in health facilities in the future (AOR = 3.17, 95% CI = 1.50–6.70; *p* = 0.002) (See Table 5)

## 4. Discussion

Most adolescents, irrespective of their socio-demographic background, engage in sexual behaviours that could expose them to sexual and reproductive health challenges, such as STIs/HIV infections and unwanted pregnancies [9]. In Nigeria, access to sexual and reproductive healthcare or the willingness to seek SRH care in health facilities when the need arises is a matter of concern as a result of the negative outcome when adolescents seek care from unqualified people, resulting in an increased burden of morbidity and mortality [16,17,18]. In light of these concerns, the current research aimed to understand if adolescents accessed SRH care, and what influenced such health-seeking behaviour. To this end, we administered a survey to adolescents in Plateau State, Nigeria, across gender, marital status, school status and educational level. 

The results showed that more than one-third of the participating adolescents were currently sexually active, usually without any form of protection such as a condom. This is comparable with what was documented from a study among adolescents with similar characteristics, reporting a high sexual activity with the risk of exposure to negative outcomes such as unintended pregnancy, HIV and STIs [19].

Despite the risk of sexual activity and the relatively high (about three-quarters) level of awareness of SRH issues, although higher for HIV than for STIs, nearly three-quarters of participants had never visited a health facility for information or services on SRH, and more than half were not willing to visit a health facility in the future for any SRH issues. Similarly, some studies in Nigeria reported good knowledge of SRH among adolescents, albeit, mostly centred on HIV and less about other STIs [23,24]. However, as we also report in the current study, despite the substantial knowledge of HIV recorded among adolescents, this has not affected their health seeking, as many of those who needed SRH services such as condoms, contraceptives and treatment of STIs did not do so in health facilities but rather sought care from non-professionals [16,24]. 

The reasons provided by adolescents in this study for not utilising, or wanting to utilise, these services in the health facilities were mainly lack of privacy and confidentiality, the negative attitude of health care providers, the cost of services and commodities that they are unable to afford, and the non-availability of the services for adolescents. We have seen from previous studies and reviews of other studies that there is inadequate privacy and confidentiality provided for adolescents in health care settings [18]. Additionally, the SRH services provided in most health facilities were adult-centred without any consideration for adolescents [25,26]. 

The unwillingness of adolescents to utilise health facilities is of concern, as young people can benefit from information and help regarding SRH issues when needed. Our results indicate that only about half of the participating adolescents would seek care in health facilities when in need, indicating that they would either self-medicate by treating themselves when they experience SRH issues or would seek unprofessional care [27]. Sexual and reproductive health care is a specialised type of care requiring trained providers, which is why it is important that care is sought from trained health care providers [27,28].

We assessed the relationship between the socio-demographic characteristics of adolescents and their SRH care seeking, and found that younger adolescents, unmarried adolescents and adolescents who were out of school were less likely to seek SRH care in a health facility. However, the multivariate analysis showed that being sexually active was the only significant factor associated with care seeking in health facilities. This may indicate their risk consciousness and perhaps reflect that adolescents who are at an increased risk of SRH challenges are more likely to access the services that may benefit them.

Many of the respondents indicated an unwillingness to seek SRH care in the health facilities but those with a higher level of awareness of HIV indicated more willingness to seek care in health facilities. Some studies have supported these findings with results reporting a better utilisation of health facilities for SRH care among adolescents with knowledge of SRH compared to those with less knowledge [29,30,31], but whether this will be actualised is uncertain, considering the earlier reasons given for not seeking care in health facilities. 

Given these challenges, overcoming the high rate of unintended pregnancy, STIs and HIV among adolescents will require interventions targeted towards addressing these problems in Plateau State and Nigeria. Therefore, we recommend a study to explore the perspective of health care providers on the provision of ASRH care services. The findings from both client and provider perspectives will strengthen informed decision making towards the most appropriate intervention to improve the utilisation of health facilities for SRH in Nigeria.

## 5. Limitations

Although the study may not be generalised to the whole of Nigeria, it can, however, be generalised to Plateau State considering the stepwise systematic approach in the sampling and selection of the respondents. These findings may not be different from what is obtainable in other parts of the country. On the other hand, while this study has tried to uncover the health-seeking behaviour and future willingness of adolescents to seek care in health facilities, the possibility of social-desirability bias in reporting health seeking in health facilities can not be ruled out, for the fear of being judged for seeking SRH care in the wrong places. This was, however, mediated by allowing respondents to privately fill out a self-administered questionnaire without interference.

## 6. Conclusions 

In conclusion, this study shows a fairly limited utilisation and willingness of adolescents to seek SRH care in health facilities, mostly for the lack of privacy and confidentiality and the negative attitude of health care providers, suggesting an opportunity to promote SRH care and address the challenge of privacy and confidentiality for adolescents in health facilities. Being sexually active was a significant covariate of seeking care in health facilities, while education and awareness of HIV were significantly associated with a willingness to seek SRH care in health facilities in the future. Therefore, there is a need for healthcare workers to consider the possibility of SRH health promotion activities for adolescents in schools and communities, ahead of becoming sexually active, in order to mitigate the negative health-seeking from unprofessional practitioners when in need. 

## Figures and Tables

**Table 1 healthcare-10-00301-t001:** Socio-demographic characteristics of adolescents.

Socio-Demographic Characteristics	Frequency	Percentage
*Gender*		
Male	222	51.9
Female	206	48.1
*Age in years*		
18	224	52.3
19	204	47.7
*Marital status*		
Married	25	5.8
Not married	403	94.2
*Religion*		
Christianity	357	83.4
Islam	71	16.6
*Schooling status*		
In school	252	58.9
Out of school	176	41.1
*Educational status*		
None/primary	50	11.7
Secondary Tertiary	317 61	74.0 14.3

**Table 2 healthcare-10-00301-t002:** Sexual activity, awareness of SRH issues, STI, HIV and health-seeking behaviour.

Variables	Frequency	Percentage
Currently sexually active		
Yes	163	38.1
No	265	61.9
Aware of SRH issues		
Yes	227	53
No	201	47
Aware of STI		
Yes	312	73
No	116	27
Aware of HIV		
Yes	363	85
No	65	15
Use health facility for SRH issues		
Yes	100	23.4
No	328	76.6
Willingness to use health facility if there was ever a problem		
Yes	188	43.9
No	240	56.1

**Table 3 healthcare-10-00301-t003:** Reasons for non-utilisation of health facilities by adolescents (multiple responses).

Reasons	Frequency (*n* = 428)	Percentage
Lack of privacy and confidentiality	283	66.1
Cost of services and commodities	167	39.0
Attitude of health care providers	292	68.2

**Table 4 healthcare-10-00301-t004:** Covariate of SRH care seeking in health facility (*n* = 428).

	Seeking Care for SRH in Health Facility							
	Yes	No	OR	95% CI	*p*	AOR	95% CI	*p*
**Gender**								
Male	49(22.1)	173(77.9)	0.86	0.55–1.35	0.512	0.83	0.50–1.39	0.488
Female	51(24.8)	155(75.2)						
**Age (yrs)**								
18	38(17.0)	186(83.0)	0.47	0.30–0.75	0.001	0.86	0.49–1.53	0.614
19	62(30.4)	142(69.6)						
**Marital status**								
Married	10(40.0)	15(60.0)	2.32	1.01–5.34	0.048	1.88	0.74–4.77	0.182
Not married	90(22.3)	313(77.7)						
**Religion**								
Christianity	89(24.9)	268(75.1)	1.81	0.91–3.60	0.090	1.33	0.61–2.94	0.474
Islam	11(15.5)	60(84.5)						
**Schooling status**								
In school	48(19.0)	204(81.0)	0.56	0.36–0.88	0.012	0.92	0.52–1.64	0.781
Out of school	52(29.5)	124(70.5)						
**Educational status**								
Non-formal	17(41.5)	24(58.5)	1.18	0.88–158	0.279	1.33	0.95–1.87	0.099
Primary	3(33.3)	6(66.7)						
Secondary	57(18.0)	260(82.0)						
Tertiary	23(37.7)	38(62.3)						
**Sexual activity**								
Yes	68(41.7)	95(58.3)	0.19	0.12–0.31	0.001	0.05	0.01–0.49	0.011 *
No	32(12.1)	233(87.9)						
**Aware of SRH**								
Yes	62(27.3)	165(72.7)	1.61	1.02–2.55	0.041	1.04	0.63–1.72	0.887
No	38(18.9)	163(81.1)						
**Aware of STI**								
Yes	87(27.9)	225(72.1)	3.06	1.64–5.74	0.001	2.10	0.98–4.48	0.056
No	13(11.2)	103(88.8)						
**Aware of HIV**								
Yes	95(26.2)	268(73.8)	4.25	1.66–10.91	0.003	2.36	0.78–7.16	0.130
No	5(7.7)	60(92.3)						

* Significant *p*-value.

**Table 5 healthcare-10-00301-t005:** Covariate of willingness to seek care in health facility (*n* = 428).

	Willingness to Seek SRH Care in Health Facility							
	Yes	No	OR	95% CI	*p*	AOR	95% CI	*p*
**Gender**								
Male	96(43.2)	126(56.8)	0.94	0.64–1.38	0.768	0.95	0.64–1.42	0.814
Female	92(44.7)	114(55.3)						
**Age (yrs)**								
18	102(45.5)	122(54.5)	1.15	0.78–1.68	0.482	1.17	0.74–1.84	0.506
19	86(42.2)	118(57.8)						
**Religion**								
Christianity	162(45.4)	195(54.6)	1.44	0.85–2.43	0.176	1.32	0.75–2.32	0.337
Islam	26(36.6)	45(63.4)						
**Schooling status**								
In school	115(45.6)	137(54.4)	1.184	0.80–1.75	0.394	1.46	0.92–2.32	0.112
Out of school	73(41.5)	103(58.5)						
**Educational status**								
Non-formal	22(53.7)	19(46.3)	1.21	0.93–1.57	0.154	1.34	1.00–1.80	0.049 *
Primary	6(66.7)	3(33.3)						
Secondary	134(42.3)	183(57.7)						
Tertiary	26(42.6)	35(57.4)						
**Sexual activity**								
Yes	80(49.1)	83(50.9)	0.71	0.48–1.06	0.452	0.69	0.39–1.24	0.218
No	108(40.8)	157(59.2)						
**Aware of SRH**								
Yes	104(45.8)	123(54.2)	1.18	0.80–1.73	0.403	0.88	0.57–1.37	0.581
No	84(41.8)	117(58.2)						
**Aware of STI**								
Yes	145(46.5)	167(53.5)	1.47	0.95–2.28	0.082	0.92	0.52–1.63	0.768
No	43(37.1)	73(62.9)						
**Aware of HIV**								
Yes	172(47.4)	191(52.6)	2.76	1.51–5.03	0.001	3.17	1.50–6.70	0.002 *
No	16(24.6)	49(75.4)						

* Significant *p*-value.

## Data Availability

The dataset(s) supporting the conclusions of this article is(are) included within the article.

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
