# Peer review of "Adolescents’ Sexual and Reproductive Healthcare-Seeking Behaviour and Service Utilisation in Plateau State, Nigeria"

_healthcare, 2022, doi:10.3390/healthcare10020301_

Round 1
Reviewer 1 Report
Dear Authors,
I read the article “Adolescents’ Sexual and Reproductive Healthcare Seeking Behaviour and Service Utilisation in Plateau State, Nigeria”. First of all, the article deals with a very current topic, above all in countries as the Nigeria. It is also updated with the most recent local literature.
I have to report some considerations:
- In the introduction of the article, you should explain the Nigerian social and religious settings and how this could influence adolescents’ sexual behaviour and their approach to health facilities. Moreover, you should describe how the health facilities work and are organised (for example, are they public or private structures?);
- Why were only 18-19 year olds included? Explain the exclusion criteria.
- Page 3 - Results section: modified “Slightly” with “slightly”
- Table 1 and line 3 of section 3, you switch percentages of adolescents in/out of school: which is the correct one? This could change the results;
- Are the questionnaires anonymous? I think it’s important to say this because it could create a bias in the adolescent’s answers;
- It could be helpful for readers to add in the article the questionnaire used;
- In the Conclusions, you speak about promotion of sexual healthcare but it’s necessary, in view of the results, to explain how.
Author Response
Dear Sir/Ma
Thank you so much your review and valuable comments.
The comments raised have been addressed. Find attached

Reviewer 2 Report
Dear Authors
Thank you a well written paper.
Few comments are in the document attached.

Author Response
Dear Sir/Ma
Thank you for your valuable comments. It has been addressed. Please find attached
